# The Impact of BMI Changes on the Incidence of Glomerular Hematuria in Korean Adults: A Retrospective Study Based on the NHIS-HEALS Cohort

**DOI:** 10.3390/biomedicines11030989

**Published:** 2023-03-22

**Authors:** Yu-Jin Kwon, Mina Kim, Hasung Kim, Jung Eun Lee

**Affiliations:** 1Department of Family Medicine, Yongin Severance Hospital, Yonsei University College of Medicine, Seoul 03722, Republic of Korea; digda3@yuhs.ac; 2Data Science Team, Hanmi Pharm. Co., Ltd., Seoul 05545, Republic of Korea; mina.kim92@hanmi.co.kr (M.K.); hasung@hanmi.co.kr (H.K.); 3Division of Nephrology, Department of Internal Medicine, Yongin Severance Hospital, Yonsei University College of Medicine, Seoul 03722, Republic of Korea

**Keywords:** body mass index, weight loss, hematuria, glomerular disease, kidney disease

## Abstract

Obesity and recurrent hematuria are known risk factors for chronic kidney disease. However, there has been controversy on the association between obesity and glomerular hematuria. This study aimed to investigate the association between body mass index (BMI) and weight change and recurrent and persistent hematuria in glomerular disease using a large-scale, population-based Korean cohort. Data were collected from the National Health Insurance Service-National Health Screening Cohort. Cox proportional hazards regression analysis was used to calculate hazard ratios (HRs) and 95% confidence intervals (CIs) for recurrent and persistent hematuria in glomerular disease according to the BMI group. Compared with the BMI 23–25 kg/m^2^ group, the HR (95% CI) for incident recurrent and persistent hematuria in glomerular disease was 0.921 (0.831–1.021) in the BMI <23 kg/m^2^ group, 0.915 (0.823–1.018) in the BMI 25–30 kg/m^2^ group, and 1.151 (0.907–1.462) in the BMI ≥30 kg/m^2^ group. Compared with the stable weight group, the HRs (95% CIs) for incident recurrent and persistent hematuria in glomerular disease were 1.364 (1.029–1.808) and 0.985 (0.733–1.325) in the significant weight loss and gain groups, respectively. Despite adjusting for confounders, this result remained significant. Baseline BMI was not associated with the risk of incident recurrent and persistent hematuria in glomerular disease. Weight loss greater than 10% was associated with the incidence of recurrent and persistent hematuria in glomerular disease. Therefore, maintaining an individual’s weight could help prevent recurrent and persistent hematuria in glomerular disease in middle-aged and older Korean adults.

## 1. Introduction

Hematuria is the presence of red blood cells (RBCs) in the urine. These RBCs originate from the glomeruli, renal tubules, interstitium, and urinary tract [1]. The presence of glomerular hematuria is a marker of glomerular filtration barrier dysfunction or damage and results in RBCs with irregular shapes in the urine [2,3]. The prevalence of glomerular hematuria in the general population ranges from 0.18% to 38.7% [3]. A Korean study using data from Korea’s national health nutrition examination survey reported that 31.8% of participants aged 10 years and older had isolated hematuria [4]. Recent research suggested that glomerular hematuria promotes oxidative stress, inflammation, and kidney injury [3]. Vivante et al. reported that persistent microscopic hematuria in young adults was significantly associated with the incidence of end-stage renal disease (ESRD) over a 22-year follow-up period [5].

IgA nephropathy (IgAN) is the most common primary glomerular disease worldwide and accounts for half of the glomerular diseases diagnosed by renal biopsy conducted after microscopic hematuria is detected via urinary screening programs [6,7]. Although hematuria is not a specific characteristic of IgAN, recurrent and persistent hematuria is one of the typical symptoms.

Several studies have reported that a high body mass index (BMI) induces enlargement and ultrastructural modification of glomeruli [8]. Obesity leads to the development of kidney disease via direct and indirect mechanisms [9]. Adiposity affects the kidney directly by altering adipokines, such as leptin, resistin, and adiponectin, increasing inflammatory cytokines; oxidative stress; insulin resistance; and activating the renin-angiotensin-aldosterone system [10]. Increased renal metabolic demands with increased body weight result in glomerular hyperfiltration, hypertrophy, and focal or segmental glomerulosclerosis [11]. Obesity also indirectly affects kidneys through the development of metabolic syndrome, diabetes, hypertension, and dyslipidemia [12]. Previous studies suggested that reducing body weight could improve the outcomes of kidney diseases [13,14]. A single-center cohort study [15] reported that low BMI is a significant risk factor for kidney disease progression in IgAN. Another study showed that weight loss is associated with worse outcomes in patients with chronic kidney diseases (CKD) [16]. However, some studies did not find any association between BMI and IgAN progression [17,18,19]. Currently, the literature shows inconsistent results on associations between BMI and IgAN progression. Furthermore, the role of body weight changes in the incidence of IgAN or glomerular disease with hematuria is still unclear. 

Thus, this study aimed to investigate the association between baseline BMI and BMI changes and the incidence of recurrent and persistent hematuria in glomerular disease. 

## 2. Materials and Methods

### 2.1. Data Source

Data were collected from the Korean National Health Insurance Service-National Health Screening Cohort (NHIS-HEALS) from 2002 to 2015 [20]. Detailed information about the NHIS-HEALS was previously published [20]. The NHIS-HEALS is the national health screening program of Korea, which is designed to prevent and detect non-communicable diseases and health risk factors. The screening program was conducted at least every 2 years among participants aged between 40 and 79 years from 2002 and followed up through 31 December 2015. The Institutional Review Board (IRB) of Yongin Severance Hospital approved the present study (IRB No: 9-2020-0009), which was conducted according to the guidelines of the Declaration of Helsinki.

### 2.2. Study Population

To diminish any potential effects, participants who underwent health screening in 2002 were excluded. The results on the presence of hematuria by urine dipstick were collected until 31 December 2008. In total, 486,055 participants who underwent their first health screening from 1 January 2003 to 31 December 2008 were included in this study. The follow-up period was from 1 January 2003 to 31 December 2015. Individuals who (1) had proteinuria/hematuria, trace or above, by urine dipstick (*n* = 59,902); (2) were diagnosed with recurrent and persistent hematuria in glomerular disease (10th edition of the International Classification of Diseases [ICD-10] codes N02.0–N02.8) (*n* = 231); (3) were diagnosed with malignant neoplasia (ICD-10 codes C), CKD, or ESRD (ICD codes N180, N181, N182, N183, N184, N185, N189) (*n* = 16,318); and (4) had missing data for age, sex, BMI, or other variables (*n* = 23,162) at the time of enrollment were excluded. Finally, 386,422 participants were included. The data management is described in Figure 1. 

### 2.3. Definition of Recurrent and Persistent Hematuria

The outcome of this study was recurrent and persistent hematuria in glomerular disease. Data of participants who were diagnosed with recurrent and persistent hematuria as those who were newly diagnosed with ICD codes N02.0–N02.8 from 1 January 2003 to 31 December 2015 were analyzed. These codes are representative codes for IgAN in Korea. The details of the ICD-codes are as follows: N02.0, recurrent and persistent hematuria with minor glomerular abnormality; N02.1, recurrent and persistent hematuria with focal and segmental glomerular lesions; N02.2, recurrent and persistent hematuria with diffuse membranous glomerulonephritis; N02.3, recurrent and persistent hematuria with diffuse mesangial proliferative glomerulonephritis; N02.4, recurrent and persistent hematuria with diffuse endocapillary proliferative glomerulonephritis; N02.5, recurrent and persistent hematuria with diffuse mesangiocapillary glomerulonephritis; N02.6, recurrent and persistent hematuria with dense deposit disease; N02.7, recurrent and persistent hematuria with diffuse crescentic glomerulonephritis; and N02.8, recurrent and persistent hematuria with other morphologic changes.

### 2.4. Classification of BMI at Baseline and Definition of BMI Changes

BMI was calculated as body weight divided by height squared (kg/m^2^). Baseline BMI was defined as BMI measured at the first health examination from 2003 to 2008. Baseline BMI was categorized into four groups according to the 2018 Korean Society for the Study of Obesity Guidelines: <23 kg/m^2^, 23 to <25 kg/m^2^, 25 to <30 kg/m^2^, and ≥30 kg/m^2^ [21]. Reference BMI was set at 23–25 kg/m^2^. 

The mean follow-up duration was 6 years. Follow-up BMI was defined as the last checked BMI within 6 years. The BMI change was calculated as the nearest BMI at the end of the follow-up minus the baseline BMI. Patients who had missing follow-up data for BMI within 6 years and those who were diagnosed with recurrent and persistent hematuria in glomerular disease within 6 years were excluded. The flow chart is presented in Appendix A. In this analysis, only the incidence of recurrent and persistent hematuria in glomerular disease that occurred after 6 years was considered. The BMI changes during the follow-up period were defined as follows: (follow-up BMI-baseline BMI) ×100/baseline BMI. The BMI changes were categorized into quartiles: <−3.33% (Q1), −3.33–0% (Q2), 0–3.59% (Q3), and ≥3.59% (Q4). 

Moreover, BMI changes were classified into three groups: weight loss (<−10%), stable weight (−10% to < 10%), and weight gain (≥10%) groups. The reference groups were set as 0–3.59% (Q3) or −10% to <10%. 

### 2.5. Variables

The variables included in the analysis were age, sex, blood pressure (BP), blood glucose level, total cholesterol level, smoking status, alcohol consumption, physical activity, hypertension, diabetes, dyslipidemia, and income status. For these variables, the values measured at the first screening were considered. For the smoking status, the participants were categorized as non-smokers, ex-smokers, or current smokers. Alcohol consumption was categorized into three groups: “rare”, less than twice per month; “sometimes”, twice per month to twice per week; and “often”, more than twice per week. Physical activity level was categorized into three groups: ‘rare’, did not exercise; ‘sometimes’, exercised 1–4 times per week; and “regular”, exercised more than five times per week. Economic status was categorized into three groups based on individual income percentile: “low”, 0–30th percentile; “middle”, 40–70th percentile; and “high”, 80–100th percentile. Hypertension was defined as ICD I10, I11, I12, I13, I15; diabetes, ICD E10–14; and dyslipidemia, ICD E78.

### 2.6. Statistical Analysis

All data are presented as the number of participants (%) or the mean ± standard deviation as appropriate for each variable. To compare the baseline characteristics across the BMI groups, a one-way analysis of variance was used for continuous variables, and the chi-squared test was used for categorical variables. A Cox proportional hazard spline plot was used to assess associations between baseline BMI and BMI changes and the incidence of recurrent and persistent hematuria. The incidence per 1000 person-years was calculated for each group. Cox proportional hazards regression models were used to determine the hazard ratio (HR) and 95% confidence interval (CI) for the incidence of recurrent and persistent hematuria in glomerular disease. An age and sex-adjusted model was set. All multivariable models were adjusted for smoking, alcohol drinking, physical activity, hypertension, diabetes, and dyslipidemia. All *p*-values were two-sided, and *p*-values < 0.05 were considered statistically significant. All statistical analyses were performed using SAS version 9.4 (SAS Institute Inc., Cary, NC, USA) and R version 3.5.2 (The R Foundation for Statistical Computing, Vienna, Austria; http://www.R-project.org/ (accessed on 28 February 2022)).

## 3. Results

Table 1 shows the demographic characteristics of the study cohort stratified by BMI category. Those with higher BMI were older, more likely to be women, and had a higher mean systolic BP, diastolic BP, fasting blood glucose, and total cholesterol. Those with lower BMI were more likely to be current smokers, consume more alcohol, and have a higher household income status.

During the mean follow-up of 6 years, there were 2254 incidents of recurrent and persistent hematuria in glomerular disease events. The incidence in the BMI < 23 kg/m^2^ group was 0.50/1000 person-years; BMI 23–25 kg/m^2^, 0.54/1000 person-years; BMI 25–30 kg/m^2^, 0.50/1000 person-years; and BMI ≥ 30 kg/m^2^, 0.63/1000 person-years. A multivariable Cox proportional hazards regression model was conducted to determine the relationship between BMI and the incidence of recurrent and persistent hematuria in glomerular disease (Table 2).

Compared to the BMI 23–25 kg/m^2^ group, the HR (95% CI) for incident recurrent and persistent hematuria in glomerular disease was 0.921 (0.831–1.021) in the BMI < 23 kg/m^2^ group; BMI 25–30 kg/m^2^ group, 0.915 (0.823–1.018); and BMI ≥ 30 kg/m^2^, 1.151 (0.907–1.462). The model was adjusted for age and sex; however, there were no significant associations between BMI and the incidence of recurrent and persistent hematuria. Furthermore, after adjusting for age, sex, smoking, alcohol drinking, physical activity, hypertension, diabetes, and dyslipidemia, the results remained insignificant. Figure 2A shows a spline curve for incident recurrent and persistent hematuria in glomerular disease according to baseline BMI.

Table 3 shows the independent relationship between BMI change quartiles and incident recurrent and persistent hematuria in glomerular disease, presented as HRs and 95% CIs. Compared to Q3 (0–3.59%), HRs (95% CI) for incident recurrent and persistent hematuria in glomerular disease were 1.204 (1.017–1.427) in Q1, 1.139 (0.955–1.360) in Q2, and 1.056 (0.887–1.257) in Q4. After adjusting for age, sex, smoking, alcohol drinking, physical activity, hypertension, diabetes, and dyslipidemia, the significant results were attenuated (Q1 (<−3.33%) vs. Q3 (0–3.59%); HR and 95% CI = 1.147 (0.968–1.359). Compared to the stable weight group (−10–10%), the HRs (95% CI) for incident recurrent and persistent hematuria in glomerular disease were 1.364 (1.029–1.808) and 0.985 (0.733–1.325) in the weight loss (<−10%) and gain groups (≥10%), respectively. The model was adjusted for the same confounders, and a significant association between weight loss and incident recurrent and persistent hematuria was observed. Figure 2B shows a spline curve for incident recurrent and persistent hematuria in glomerular disease according to BMI changes. Greater weight loss was linearly associated with incident recurrent and persistent hematuria in glomerular disease.

## 4. Discussion

Although hematuria has been considered a representative symptom of renal disease, its role in the progression of kidney disease has received less attention than that given to proteinuria [22]. Recent studies have suggested that glomerular hematuria has a pathological role in promoting acute kidney injury (AKI) and progression to CKD through its nephrotoxic action of hemoglobin, heme inflammation, and oxidative stress [22,23]. Several epidemiological studies have reported that persistent hematuria is associated with an increased risk of renal function decline and ESRD during long follow-up periods [5,24,25]. Considering the important clinical implications of hematuria, it is important to investigate the effects of BMI on recurrent and persistent hematuria in glomerular disease. However, there are still conflicting results about the association between BMI and glomerular hematuria. Further large-scale epidemiologic studies are required to clarify the association between BMI and the incidence of recurrent and persistent hematuria in glomerular disease. 

This study investigated the association between BMI and recurrent and persistent hematuria in glomerular disease diagnosed with ICD-10 codes recorded directly by clinicians using the large, population-based Korean National Health Insurance Service-National Health Screening Cohort. This is the first study to investigate the null association between BMI and recurrent and persistent hematuria in glomerular disease using a large-scale cohort. Particularly, it was found that weight loss greater than 10% was associated with recurrent and persistent hematuria.

Experimental evidence suggests that obesity has a role in the formation and progression of some glomerular lesions [26,27], but data for human glomerulonephritis are still insufficient. Previous studies have shown that a BMI ≥25 kg/m^2^ was a risk factor for IgAN progression [28,29]. A study with 43 IgAN patients reported that a BMI ≥25 kg/m^2^ was a significant predictor of disease progression [28]. Another study with 193 Japanese IgAN patients showed that even a slightly high BMI is a risk factor for disease progression [29]. In a recent cross-sectional study with 537 patients [30], the obese group was significantly associated with higher mesangial matrix expansion scores compared to the normal weight/overweight group (*p* = 0.020). In a cohort of 331 patients with biopsy-proven IgAN [17], patients with BMI greater than 25 kg/m^2^ had worse clinical outcomes; however, there was no direct association between BMI and IgA progression. The studies were conducted on patients who had pre-existing IgAN, while this study investigated the effect of BMI on new-onset IgAN in the general population. 

Unlike previous studies, this study found that BMI status was not associated with the incidence of recurrent and persistent hematuria in glomerular disease. In the current study, BMI was divided into four groups (<23 kg/m^2^, 23–25 kg/m^2^, 25–30 kg/m^2^, and ≥30 kg/m^2^). There was no difference in the results when further analysis was conducted by dividing the BMI into various criteria, such as BMI 18.5 kg/m^2^ (cut-off). One of the possible reasons was the attributes of BMI. Traditionally, BMI is an indicator of obesity, but this study was unable to distinguish between body fat and muscle mass [31]. Additionally, BMI could not represent regional fat distribution. Further, obesity itself may affect the prevalence of glomerulopathy [32], exacerbating IgAN that has already occurred; however, obesity may have no impact on the incidence of hematuria in glomerular disease. More studies are needed using other indicators, such as body fat and waist circumference. In the current study, data on waist circumference were missing, and, therefore, it was excluded from further analysis.

Although BMI has limitations in indicating body components, it is commonly used as a marker in the assessment of nutritional status [33]. Several studies have reported the obesity paradox in populations with chronic heart failure [34], chronic obstructive pulmonary disease [35], and CKD [36,37]. The obesity paradox is likely to be explained by the fact that weight loss and physical frailty are associated with mortality in patients with chronic heart failure, chronic lung disease, and CKD [37]. The Chronic Renal Insufficiency Cohort Study, a multicenter prospective cohort study, showed that weight loss was associated with a 54% higher risk of death after dialysis therapy initiation in the stable weight group [16]. Ouyang et al. [15] showed that the incidence of ESRD in the underweight (<18.5 kg/m^2^) group was higher than that in other BMI groups during a 47.1-month follow-up period. Unintentional weight loss, which might reflect nutritional deficit, has been reported as being associated with the progression of CKD [38]. Similar results were observed in the current study. This study found that the reduced BMI group (Q1 <−3.33% during the follow-up period) was associated with an increased risk of recurrent and persistent hematuria in glomerular disease. This association was attenuated after adjusting for confounding variables. However, interestingly, significant weight loss (greater than 10% during the follow-up period) was associated with an increased risk of recurrent and persistent hematuria in glomerular disease. The result remained significant even after adjusting for the same confounders. 

This study has several limitations. First, the major limitation was that renal biopsy results were unobtainable. Since data from the NHIS-HEALS were used, recurrent and persistent hematuria in glomerular disease was defined using ICD-10 codes (N02.0–N02.8). Therefore, it was not possible to determine whether physicians recorded diagnostic codes based on renal biopsy results. According to a previous study conducted in Korea, among patients with asymptomatic isolated microscopic hematuria, IgAN was the most common cause, followed by idiopathic mesangial proliferative glomerulonephritis, with prevalence rates of 46.9% and 43.1%, respectively. IgAN, the most common cause of recurrent hematuria, is usually recorded in young adults during school urine screening tests [7]. However, the participants from NHIS-HEALS were middle-aged and older adults. Therefore, the incidence of IgAN could be lower than that previously reported [39]. Second, diseases commonly associated with recurrent hematuria, such as inflammatory conditions of the urethra, bladder, and prostate and malignancies that developed during the follow-up period, were not completely ruled out. Third, although we found that weight loss was associated with the incidence of recurrent and persistent hematuria, we could not determine the intentionality of weight loss in the cohort study. Unintentional weight loss is an involuntary decline in body weight over time and could reflect chronic diseases, psychological factors, social conditions, and undiagnosed illnesses [40]. In addition, only the association between BMI changes within 6 years and incident recurrent and persistent hematuria in glomerular disease was observed. Annual BMI changes could be a better predictor for incident recurrent and persistent hematuria in glomerular disease. Fourth, adiposity itself, directly and indirectly, affects the kidney through its endocrine activity, such as the production of adiponectin, leptin, and other inflammatory cytokines [5]. However, BMI could not fully reflect the amount of adiposity because the definition of BMI did not distinguish muscle from fat mass. Finally, this study included middle-aged and older Korean adults; therefore, there could have been a selection bias. Therefore, the results from this study cannot be generalized to other races/ethnicities. Further research is required to establish the association between body weight and recurrent and persistent hematuria in glomerular disease by considering the body composition data for young adults and annual changes in BMI. Finally, angiotensin-converting enzyme inhibitors/angiotensin receptor blockers are widely used in hypertension and kidney disease. However, the effect of these drugs could not be determined. Despite these weaknesses, this is the first study to investigate the impact of BMI and BMI changes on recurrent and persistent hematuria in glomerular disease using large-scale, population-based data. Although the results based on renal biopsy could not be determined, ICD-10 codes were used, which the physicians directly coded during the medical care process.

## 5. Conclusions

Baseline BMI was not associated with the incidence of recurrent and persistent hematuria in glomerular disease. Significant weight loss greater than 10% was associated with the incidence of recurrent and persistent hematuria in glomerular disease. Maintaining individuals’ BMI could be a preventive strategy for the incidence of recurrent and persistent hematuria in glomerular disease in middle-aged and older Korean adults. Moreover, studies based on renal biopsy are required to determine these associations.

## Figures and Tables

**Figure 1 biomedicines-11-00989-f001:**
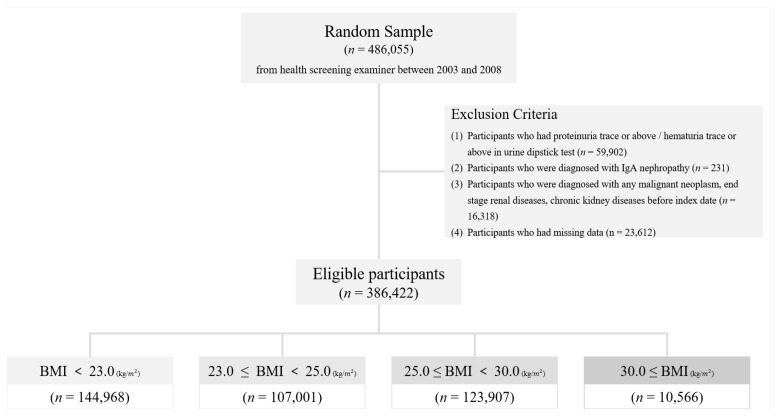
Flow chart of study population.

**Figure 2 biomedicines-11-00989-f002:**
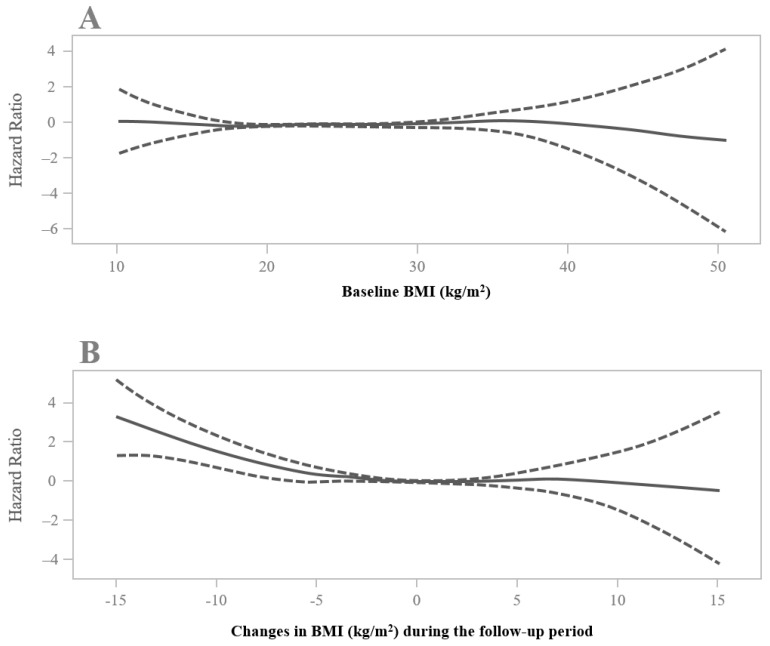
Cox proportional hazard spline curves showing associations between recurrent and persistent hematuria in glomerular disease and (**A**) baseline BMI (kg/m^2^) and (**B**) changes in BMI (kg/m^2^) during the follow-up period. BMI, body mass index.

**Table 1 biomedicines-11-00989-t001:** Baseline characteristics of the study population according to baseline BMI criteria.

	Baseline Body Mass Index				
	<23.0 kg/m^2^	23.0–25.0 kg/m^2^	25.0–30.0 kg/m^2^	≥30.0 kg/m^2^	*p*
N	144,968	107,001	123,907	10,566	<0.001
Male, *n* (%)	78,341 (54.0)	62,566 (58.5)	72,201 (58.3)	4403 (41.7)	<0.001
Age, years	53.8 ± 10.0	53.5 ± 9.2	53.8 ± 9.0	54.3 ± 9.1	<0.001
BMI, kg/m^2^	21.1 ± 1.4	24.0 ± 0.6	26.7 ± 1.3	31.8 ± 2.7	<0.001
SBP, mmHg	123.2 ± 17.5	123.4 ± 17.4	130.6 ± 17.4	135.5 ± 18.0	<0.001
DBP, mmHg	77.0 ± 11.2	77.1 ± 11.2	81.7 ± 11.3	84.3 ± 11.7	<0.001
Glucose, mg/dL	95.2 ± 30.3	95.6 ± 30.1	100.5 ± 32.0	104.8 ± 34.9	<0.001
Cholesterol, mg/dL	193.2 ± 36.3	193.8 ± 36.3	205.6 ± 37.8	209.9 ± 38.0	<0.001
Smoking status, *n* (%)					<0.001
Non-smoker	96,973 (66.9)	71,537 (66.9)	83,662 (67.5)	8044 (76.1)	
Ex-smoker	11,056 (7.6)	10,527 (9.8)	12,800 (10.3)	780 (7.4)	
Current smoker	37,539 (25.9)	24,937 (23.3)	27,445 (22.1)	1742 (16.5)	
Alcohol drinking, *n* (%)					<0.001
Rare	84,398 (58.2)	58,848 (55.0)	67,889 (54.8)	6718 (63.6)	
Sometimes	43,726 (30.2)	35,885 (33.5)	41,466 (33.5)	2905 (27.5)	
Often	16,844 (11.6)	12,268 (11.5)	14,552 (11.7)	943 (8.9)	
Physical activity, *n* (%)					<0.001
Rare	83,805 (57.8)	55,438 (51.8)	64,800 (52.3)	6245 (59.1)	
Sometimes	47,864 (33.0)	40,175 (37.5)	45,723 (36.9)	3255 (30.8)	
Regular	13,299 (9.2)	11,388 (10.6)	13,381 (10.8)	1066 (10.1)	
Income status, *n* (%)					<0.001
Low	33,995 (23.5)	22,981 (21.5)	26,689 (21.5)	2669 (25.3)	
Middle	47,684 (32.9)	33,438 (31.3)	40,086 (32.4)	3759 (35.6)	
High	63,289 (43.7)	50,582 (47.3)	57,132 (46.1)	4138 (39.2)	
Comorbidities, *n* (%)					<0.001
Hypertension	22,629 (15.6)	23,270 (21.7)	35,900 (29.0)	4455 (42.2)	
Diabetes	15,649 (10.8)	14,349 (13.4)	19,586 (15.8)	2301 (21.8)	
Dyslipidemia	14,844 (10.2)	14,805 (13.8)	21,260 (17.2)	2307 (21.8)	

BMI, body mass index; SBP, systolic BP; DBP, diastolic BP.

**Table 2 biomedicines-11-00989-t002:** Hazard ratios (HRs) and 95% confidence intervals (CIs) for the incidence of recurrent and persistent hematuria in glomerular disease according to body mass index (BMI) status.

	Baseline BMI			
	<23.0 kg/m^2^	23.0–25.0 kg/m^2^	25.0–30.0 kg/m^2^	≥30.0 kg/m^2^
Total No.	144,968	107,001	123,907	10,566
Event	814	662	703	75
Person-years	1,626,690.04	1,217,843.1	1,412,926.53	119,835.45
IR (per 1000)	0.50	0.54	0.50	0.63
Crude HR	0.921 (0.831–1.021)	1.00 (reference)	0.915 (0.823–1.018)	1.151 (0.907–1.462)
Age- and sex-adjusted HR	0.913 (0.824–1.012)	1.00 (reference)	0.907 (0.816–1.009)	1.10 (0.867–1.398)
Fully adjusted HR	0.946 (0.853–1.049)	1.00 (reference)	0.885 (0.795–0.985)	1.033 (0.812–1.315)

BMI, body mass index; IR, incident rate; HR, hazard ratio; 95% CI, 95% confidence interval. Fully adjusted model: adjusted for age, sex, smoking, alcohol drinking, physical activity, hypertension, diabetes, and dyslipidemia.

**Table 3 biomedicines-11-00989-t003:** Hazard ratios (HRs) and 95% confidence intervals (CIs) for the incidence of recurrent and persistent hematuria in glomerular disease according to body mass index (BMI) changes during the follow-up period.

	Body Mass Index Changes					
	Q1 (<−3.33%)	Q2 (−3.33–0%)		Q3 (0–3.59%)		Q4 (≥3.59%)
Total No.	87,100	76,762		97,384		87,051
Event	275	233		261		246
Person-years	991,691.5	881,718.5		1,122,379.1		1,002,816.5
IR (per 1000)	0.28	0.26		0.23		0.25
Crude HR	1.204 (1.017–1.427)	1.139 (0.955–1.360)		1.00 (reference)		1.056 (0.887–1.257)
Age- and sex-adjusted HR	1.148 (1.001–1.361)	1.124 (0.942–1.341)		1.00 (reference)		1.046 (0.879–1.245)
Fully adjusted HR	1.147 (0.968–1.359)	1.123 (0.941–1.341)		1.00 (reference)		1.051 (0.883–1.251)
	Body mass index changes					
	<−10%		−10.0–10%		≥10%	
Total No.	13,759		318,304		16,234	
Event	51		918		46	
Person-years	153,186.61		3,659,063.74		186,355.18	
IR (per 1000)	0.33		0.25		0.25	
Crude HR	1.364 (1.029–1.808)		1.00 (reference)		0.985 (0.733–1.325)	
Age- and sex-adjusted HR	1.246 (1.006–1.653)		1.00 (reference)		0.960 (0.714–1.290)	
Fully adjusted HR	1.247 (1.001–1.655)		1.00 (reference)		0.967 (0.939–1.655)	

BMI, body mass index; IR, incident rate; HR, hazard ratio; 95% CI, 95% confidence interval. Fully adjusted model: adjusted for age, sex, smoking, alcohol drinking, physical activity, hypertension, diabetes, and dyslipidemia.

## Data Availability

The data presented in this study are openly available on the National Health Insurance Data Sharing Service homepage; http://nhiss.or.kr (accessed on 21 January 2021).

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
