# Peer review of "The Impact of BMI Changes on the Incidence of Glomerular Hematuria in Korean Adults: A Retrospective Study Based on the NHIS-HEALS Cohort"

_biomedicines, 2023, doi:10.3390/biomedicines11030989_

Round 1

Reviewer 1 Report

This population-based retrospective study suggested weight loss elicits an increased risk of IgA nephropathy or related diseases. The finding is interesting and the methods are generally sound. Some issues see below. 

1. The observations of BMI change and outcomes measures seem overlapping. Please make sure that all BMI changes were observed before the measure of outcome. 

2. The calculation of BMI change is problematic, which should consider the observational duration. Annual change of BMI is better. 

3. Adjusting age and sex seems inadequate. It is important to adjust more potential confounders including diabetes. 

4. Some sensitivity analyses are necessary to confirm the robustness of the findings including the observation period of the BMI change, the duration between last observation to the first events, different definition of the outcome and so on. 

Reviewer 2 Report

1. The title is too long and needs revision. Suggested: The impact of BMI changes on the incidence of glomerular hematuria in Korean adults: A retrospective analysis of the NHIS-HEALS data

2. The abstract requires certain improvements as well. Please include at least one sentence stating the study rationale (should be the first sentence of the abstract). 

3. The abstarct as well as the whole text should be written from a third-person point of view.

4. The overall manuscript style should be improved to meet the academic style appropriate for research manuscripts. 

5. The introduction part gives some understanding of the study rationale. However, it is narrow; please rethink and rewrite the introduction part, as the writing style should be improved. Include epidemiological data on the condition you are investigating (worldwide and local one).

6. The methods section and the results are interesting and well-written. 

7. The discussion part needs improvement.

In general, the discussion part is interesting,  however, requires improvement to meet the style appropriate for research manuscripts. The first paragraph of the discussion should provide to a potential reader the study rationale in brief.  Please find below the suggestions how the discussion part should be restructured:

Discussion

1.1 Rationale of the study (why it was done)

1.1.1 Main findings of the study

1.1.2 What makes your study unique

1.1.3 What it adds to what we already know

1.2 Subject of the discussion

Comparison of your results with neighboring countries, with countries of the same

development levels (income), with developed high-income countries). Agreement and disagreement with the studies compared

1.3 Sum up of the study, study strengths and limitations

8. The authors performed a very interesting study, and thus the conclusion should be elaborated in more details.

General comments

Please provide a point-by-point response to each of the comments and highlight the changes in the revised manuscript.

 Check the text for grammar and spelling mistakes as there are a lot of them.

Please provide a thorough point-by-point response to all comments of the reviewers and the Editor. Responses like “Done”, “ Revised”, etc. will not be accepted.

Round 2

Reviewer 1 Report

Thanks for your response. I understand the remote work on the dataset, but it does not make sense that you are not able to reanalyse the data any more.